# microRNA-mediated regulation of microRNA machinery controls cell fate decisions

**Qiuying Liu, Mariah K Novak, Rachel M Pepin, Taylor Eich, Wenqian Hu\***

Department of Biochemistry and Molecular Biology, Mayo Clinic, Rochester, United States

**Abstract** microRNAs associate with Argonaute proteins, forming the microRNA-induced silencing complex (miRISC), to repress target gene expression post-transcriptionally. Although microRNAs are critical regulators in mammalian cell differentiation, our understanding of how microRNA machinery, such as the miRISC, are regulated during development is still limited. We previously showed that repressing the production of one Argonaute protein, Ago2, by Trim71 is important for mouse embryonic stem cells (mESCs) self-renewal (Liu et al., 2021). Here, we show that among the four Argonaute proteins in mammals, Ago2 is the major developmentally regulated Argonaute protein in mESCs. Moreover, in pluripotency, besides the Trim71-mediated regulation of *Ago2* (Liu et al., 2021), *Mir182/Mir183* also repress *Ago2*. Specific inhibition of this microRNA-mediated repression results in stemness defects and accelerated differentiation through the let-7 microRNA pathway. These results reveal a microRNA-mediated regulatory circuit on microRNA machinery that is critical to maintaining pluripotency.

**\*For correspondence:**
hu.wenqian@mayo.edu

**Competing interest:** The authors declare that no competing interests exist.

## Introduction

microRNAs (miRNAs) are endogenous ~22 nucleotide (nt) RNAs with critical roles in modulating gene expression under diverse biological contexts (*Bartel, 2009*; *Bartel, 2018*). Most miRNAs are produced from long primary transcripts (pri-miRNAs) through successive processing by two double-stranded RNA (dsRNA) endonucleases named Drosha and Dicer, generating pre-miRNAs and ~22 nt dsRNAs, respectively. One RNA strand in the ~22 nt dsRNA, the mature miRNA, is selectively incorporated into the Argonaute (Ago) protein, forming the miRNA-induced silencing complex (miRISC) (*Ha and Kim, 2014*). In animals, miRISC recognizes its target mRNAs through partial base pairings mediated by the miRNA (*Bartel, 2009*). The Ago protein recruits GW182 proteins to down-regulate target mRNA expression through mRNA degradation and/or translational repression (*Nilsen, 2007*). Although miRNAs play critical regulatory roles in mammalian cell differentiation (*Ameres and Zamore, 2013*; *Ebert and Sharp, 2012*), our understanding on how miRNA machinery, particularly the miRISC, are regulated during development is still limited.

We recently found that Ago2, a key component in the miRISC, is repressed at the mRNA translation level by an RNA-binding protein named Trim71 in mouse embryonic stem cells (mESCs) (*Liu et al., 2021*). This repression of *Ago2* inhibits stem cell differentiation mediated by the conserved pro-differentiation let-7 miRNAs (*Büssing et al., 2008*; *Liu et al., 2021*). These results suggest that *Ago2* is developmentally regulated during stem cell self-renewal and differentiation, and beg for characterization of additional regulators of *Ago2*. Moreover, besides Ago2, there are three additional Ago proteins (Ago1, Ago3, Ago4) in mammals that function redundantly in the miRNA pathway (*Meister, 2013*). The relative abundance of these Ago proteins and their contribution to miRNA activities during cell differentiation, however, are still unknown.

Here, using mESC fate decisions between pluripotency and differentiation as a mammalian cell differentiation model, we determined that Ago2 is the predominant Ago protein in mESCs, and Ago2 level increases when mESCs exit pluripotency. In the pluripotent state, *Mir182* and *Mir183*, two conserved miRNAs abundantly expressed in mESCs, repress *Ago2* and control the stemness of mESCs. Specific inhibition of *Mir182/Mir183*-mediated repression of *Ago2* results in stemness defects and accelerated differentiation of mESCs through the let-7 miRNA pathway. Collectively, these results reveal an miRNA-mediated regulatory circuit on the miRNA machinery that is critical to maintaining pluripotency.

## Results

### Ago2 is the predominant developmentally regulated Ago protein in mESCs

Mammals have four Ago proteins (Ago1–4) that function redundantly in miRNA-mediated regulations (*Meister, 2013*). Transcriptomic profiling on mESCs from different laboratories indicated that mESCs express only *Ago1* and *Ago2* (*Figure 1—figure supplement 1A*; *Liu et al., 2021*; *Marks et al., 2012*). To examine the relative abundance of Ago1 and Ago2 at the protein level, we generated mESCs with a Flag-tag knocked-in at the N-terminus of the Ago1 and Ago2 loci, respectively, via CRISPR/Cas9-mediated genome editing (*Figure 1—figure supplement 1B, C*). These mESCs with the Flag-tag knocked-in displayed no stemness defects compared to the wild-type (WT) mESCs (*Figure 1—figure supplement 1D*) and enabled us to use the same antibody (e.g., anti-Flag) to compare the relative abundance of Ago1 and Ago2. Western blotting via an anti-Flag antibody indicated that Ago2 is the predominant Ago protein in mESCs at the protein level (*Figure 1A*).

To examine whether Ago2 level is regulated during mESCs differentiation, we cultured mESCs under three different conditions that mimic three different developmental stages: ground/naive state (in 2i + Lif), primed state (in 15% FBS+ Lif), and differentiating state (in 15% FBS without Lif), which resulted in decreasing stemness in mESCs, as determined by the colony formation assay (*Figure 1B*). Western blotting indicated that Ago2 level increased when mESCs exited pluripotency (*Figure 1C*). This result indicated that Ago2 is developmentally regulated in mESCs, and Ago2 level is repressed in the pluripotent state.

### *Mir182/Mir183* regulate *Ago2* and maintain stemness in mESCs

To determine how *Ago2* is regulated in mESCs, we hypothesized that miRNAs expressed in mESCs might contribute to the repression of *Ago2* because miRNAs are important negative regulators of gene expression. We identified the conserved miRNA-binding sites in the 3'UTR of *Ago2* mRNA through TargetScan (*Agarwal et al., 2015*) and then examined the expression level of the corresponding miRNAs in mESCs using existing small-RNA-seq datasets (*Liu et al., 2021 Figure 1D*). This analysis revealed that among the miRNAs that can potentially regulate *Ago2*, *Mir182*, and *Mir183*, two miRNAs from the same miRNA family that are abundantly expressed in stem cells *Dambal et al., 2015*, have significantly higher expression levels (*Figure 1E*). Interestingly, *Mir182/Mir183* decrease when mESCs transition from the ground state to the primed and differentiating state (*Hadjimichael et al., 2016*; *Wang et al., 2017*), which negatively correlates with the Ago2 expression pattern during this transition (*Figure 1C*). These observations suggest that *Ago2* is repressed by *Mir182/Mir183* in mESCs. Consistent with this notion, using RNA antisense purification, we found that *Mir182* and *Mir183* specifically associated with *Ago2* mRNA in mESCs (*Figure 1—figure supplement 2*).

Two lines of evidence indicated that *Mir182/Mir183* regulate *Ago2* mRNA. First, Ago2 increased when *Mir182*, *Mir183*, or both *Mir182* and *Mir183* were knocked out in mESCs (*Figure 2—figure supplement 1A*, *Figure 2A and B*). Second, when either *Mir182* or *Mir183* was over-expressed in the WT mESCs (*Figure 2—figure supplement 1B*), the Ago2 level decreased (*Figure 2—figure supplement 1C*). The results from these loss-of-function and gain-of-function experiments argue that *Mir182/Mir183* repress *Ago2* expression in mESCs.

Interestingly, *Mir182Δ*, *Mir183Δ*, and *Mir182Δ/Mir183Δ* mESCs displayed defects in self-renewal (*Figure 2C*), as determined by the colony formation assay in the 15% FBS + Lif medium, where differentiation was not blocked by the two inhibitors in the 2i + Lif medium. Moreover, these miRNA knockout mESCs had accelerated differentiation, as revealed by the exit pluripotency assay (*Figure 2D*), which

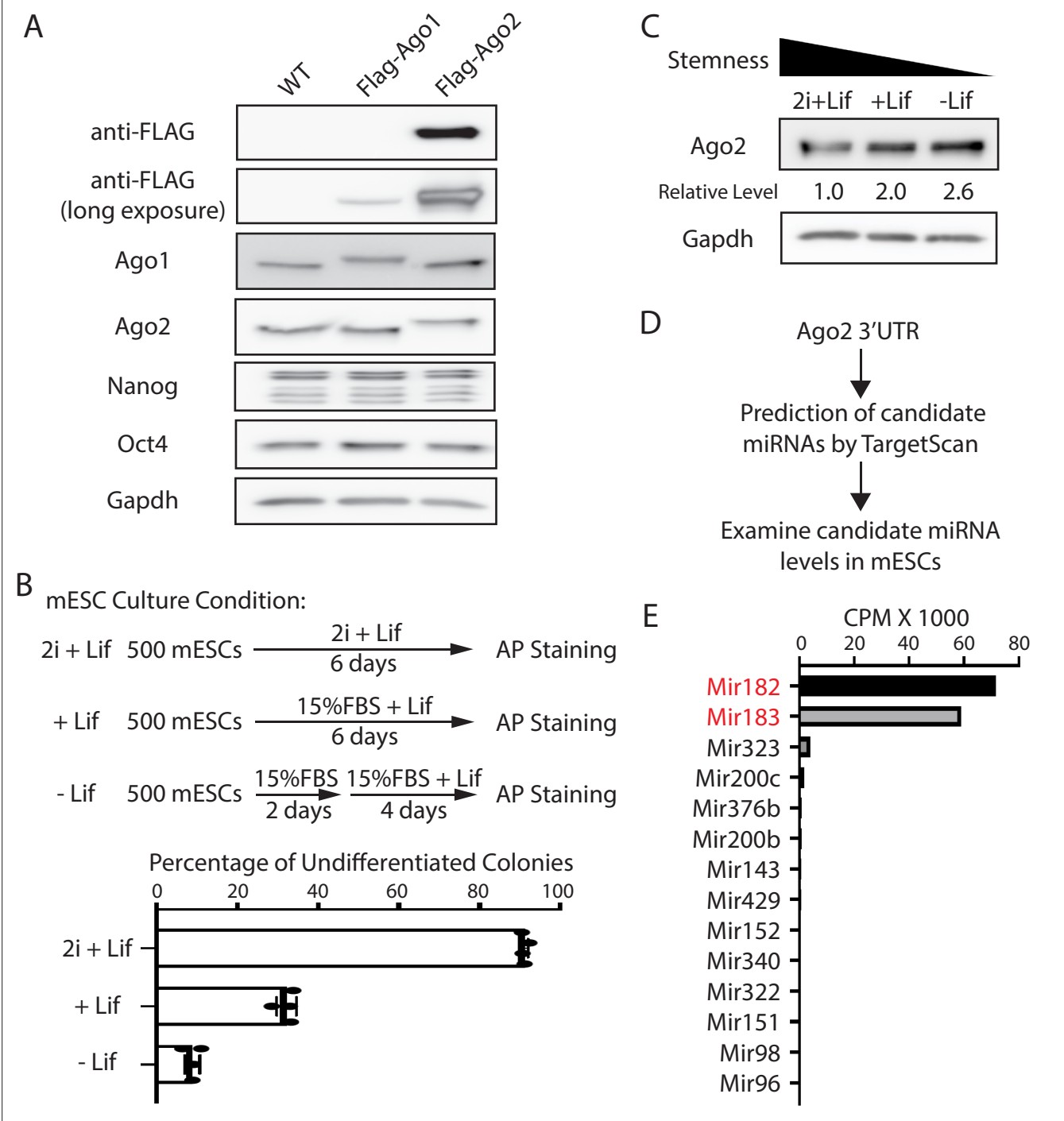

**Figure 1.** Ago2 is the major developmentally regulated Argonaute protein in mouse embryonic stem cells (mESCs). (**A**) Western blotting in the wild-type (WT), Flag-Ago1, and Flag-Ago2 mESCs. (**B**) Colony formation assay for the mESCs. The WT mESCs were cultured under the indicated conditions, and the resultant colonies were fixed and stained for AP (alkaline phosphatase activity). The results represent the means (± SD) of four independent experiments. (**C**) Western blotting in the WT mESCs cultured under the indicated conditions. (**D**) Outline of identifying miRNAs that can potentially regulate *Ago2*. (**E**) Expression levels of the identified miRNAs from (**D**) in mESCs. CPM: counts per million reads.

The online version of this article includes the following figure supplement(s) for figure 1:

**Source data 1.** Tiff files of raw gel images for *Figure 1A and C*; *Figure 1—figure supplement 1C*.

**Figure supplement 1.** Expression of Argonaute proteins in mouse embryonic stem cells (mESCs).

**Figure supplement 2.** *Mir182* and *Mir183* are associated with *Ago2* mRNA in mouse embryonic stem cells (mESCs).

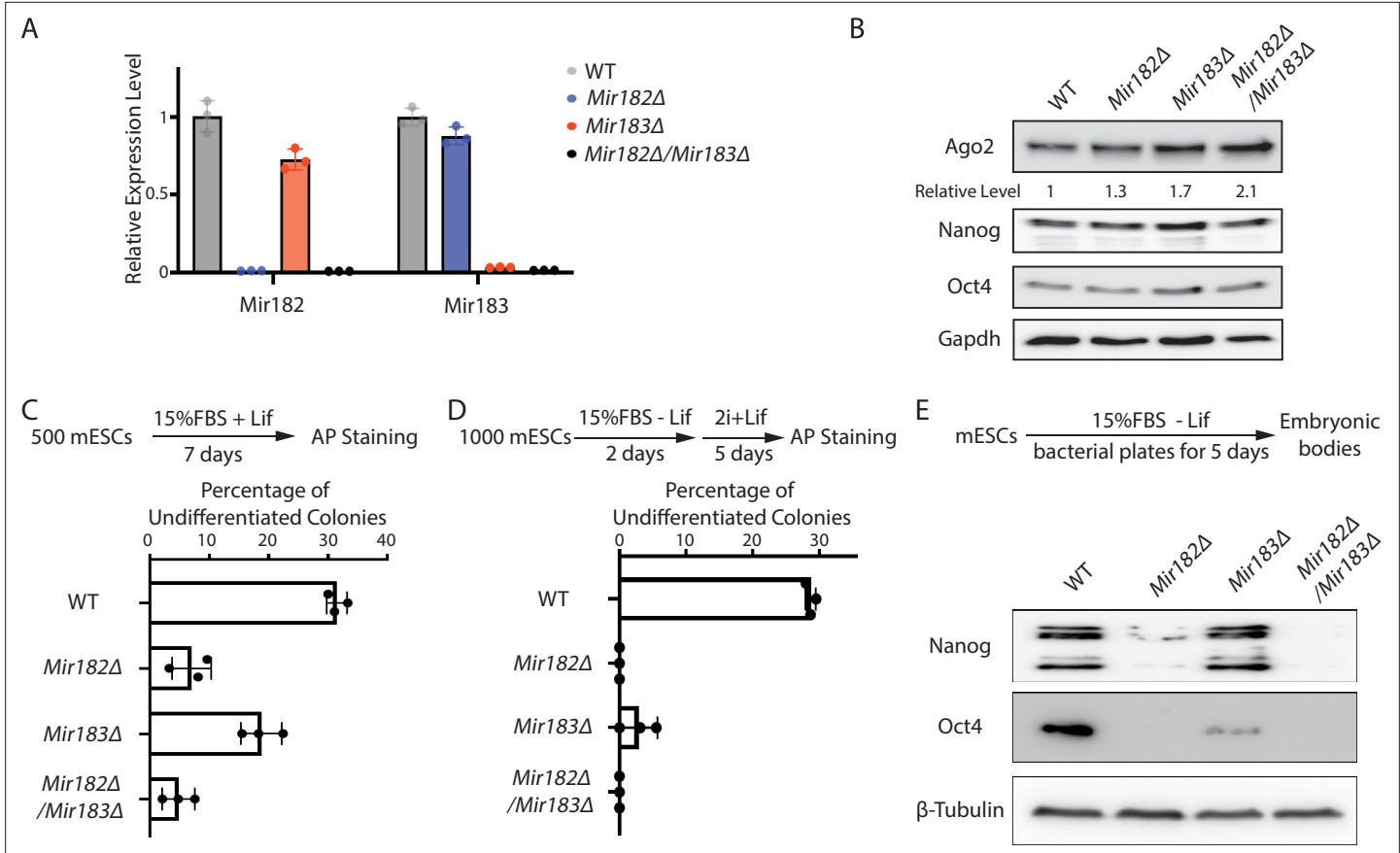

**Figure 2.** *Mir182/Mir183* regulate *Ago2* and maintain stemness in mouse embryonic stem cells (mESCs). (**A**) qRT-PCR on *Mir182* and *Mir183*. For each miRNA, the expression level in wild-type (WT) cells was set as one for relative comparison. U6 RNA was used for normalization. The results represent the means (± SD) of three independent replicates. (**B**) Western blotting in the WT, *Mir182Δ*, *Mir183Δ*, and *Mir182Δ/Mir183Δ* mESCs. GAPDH was used for normalization in calculating the relative expression levels. (**C**) Colony formation assay for mESCs. The mESCs were cultured in 15% FBS+ Lif for 7 days, and the resultant colonies were fixed and stained for alkaline phosphatase (AP). (**D**) Exit pluripotency assay for mESCs. The mESCs were induced to exit pluripotency in medium without Lif for 2 days and then switched to 2i + Lif medium for 5 days. The resultant colonies were fixed and stained for AP. In (C and D), the colony morphology and AP intensity were evaluated through microscopy; 100–200 colonies were examined each time to determine the percentage of undifferentiated colonies. The results represent the means (± SD) of three independent experiments. (**E**) Western blotting of pluripotency factors during embryoid body (EB) formation.

The online version of this article includes the following figure supplement(s) for figure 2:

**Source data 1.** Tiff files of raw gel images for *Figure 2B and E*; *Figure 2—figure supplement 1C*.

**Figure supplement 1.** *Ago2* mRNA is a target of *Mir182* and *Mir183* in mouse embryonic stem cells (mESCs).

evaluates the rate ESCs exit the pluripotent state (*Betschinger et al., 2013*), and by the measurement of pluripotency factors through Western blotting on differentiating embryonic bodies (*Figure 2E*). These cellular phenotypes suggest that *Mir182/Mir183*-mediated regulation of *Ago2* is important to mESCs.

### *Mir182/Mir183*-mediated repression of *Ago2* is required for maintaining pluripotency

A caveat in interpreting results from miRNA knockout and over-expression experiments is the pleiotropic effects. Because each miRNA can regulate hundreds of mRNAs, when an miRNA is knocked out or over-expressed, hundreds of miRNA:mRNA interactions are altered, making it difficult to determine whether a specific miRNA:mRNA interaction contributes to the phenotypical changes.

To address this issue and specifically examine the functional significance of *Mir182/Mir183*-mediated regulation of *Ago2* in mESCs, we mutated the *Mir182/Mir183*-binding sites in the 3'UTR of *Ago2* mRNA via CRISPR/Cas9-mediated genome editing (*Figure 3A and B*). Two observations

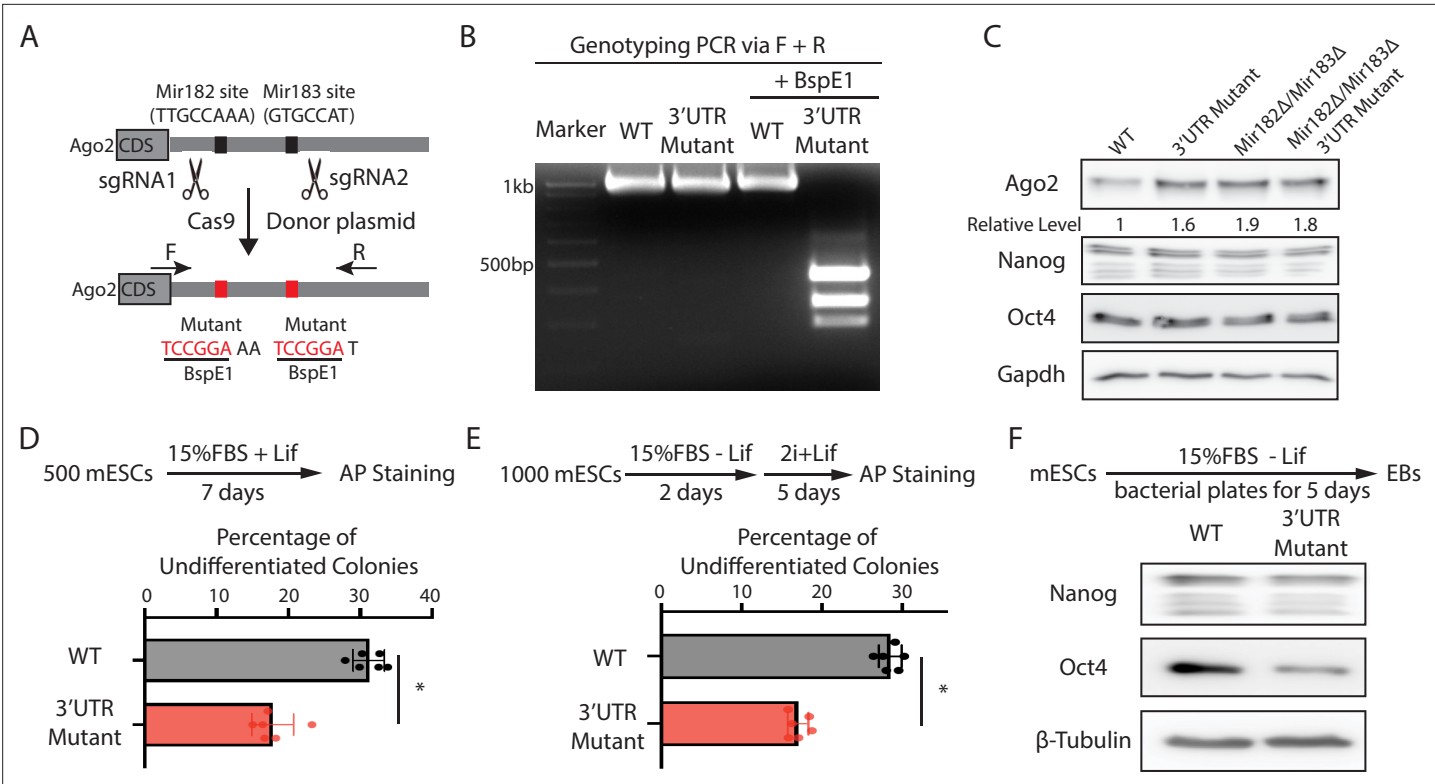

**Figure 3.** *Mir182/Mir183*-mediated repression of *Ago2* is required for maintaining pluripotency. (**A**) Mutating *Mir182*- and *Mir183*-binding sites in *Ago2* mRNA's 3'UTR via genome editing. (**B**) Genotyping of the Ago2 3'UTR mutant. The PCR was performed using the oligos (F and R) indicated in (A). (**C**) Western blotting in the wild-type (WT), *Ago2* 3'UTR mutant, *Mir182Δ/ Mir183Δ*, and *Mir182Δ/ Mir183Δ/Ago2* 3'UTR mutant. (**D**) Colony formation assay for mouse embryonic stem cells (mESCs). (**E**) Exit pluripotency assay for mESCs. In (D and E), the colony morphology and alkaline phosphatase (AP) intensity were evaluated through microscopy. The results represent the means (± SD) of four independent experiments. *p < 0.05 by the Student's t-test. Western blotting of pluripotency factors in day 5 embryoid bodies (EBs).

The online version of this article includes the following figure supplement(s) for figure 3:

**Source data 1.** Tiff files of raw gel images for *Figure 3C and F*; *Figure 3—figure supplement 1B*.

**Figure supplement 1.** Inhibition of *Mir182/Mir183*-mediated regulation of *Ago2* in mouse embryonic stem cells (mESCs).

indicated that the mutations disrupted the interaction between *Ago2* mRNA and *Mir182/Mir183*. First, similar to the miRNA knockout mESCs (*Figure 2B*), Ago2 increased in the 3'UTR mutant mESCs (*Figure 3C*). Second, in contrast to the results in the WT mESCs (*Figure 2—figure supplement 1C*), over-expression of either *Mir182* or *Mir183* in the 3'UTR mutant mESCs did not decrease Ago2 (*Figure 3—figure supplement 1A, B*). Notably, in the *Mir182Δ/Mir183Δ* mESCs, these mutations did not increase Ago2 (*Figure 3C*), indicating the increased Ago2 from these mutations in the WT mESCs is dependent on *Mir182/Mir183*. Moreover, the 3'UTR mutations did not significantly alter the *Mir182/Mir183* levels in mESCs (*Figure 3—figure supplement 1C*). Altogether, these observations indicated that the functional significance of *Mir182/Mir183*-mediated repression of Ago2 could be specifically evaluated in the 3'UTR mutant mESCs.

When subject to the colony formation assay, the 3'UTR mutant mESCs displayed a defect in maintaining undifferentiated colonies (*Figure 3D*), indicating compromised self-renewal. When differentiation was evaluated by the exit pluripotency assay, the 3'UTR mutant mESCs had an increased differentiation rate (*Figure 3E*). Consistent with these findings, differentiating embryonic bodies from the 3'UTR mutant mESCs had a lower amount of pluripotency factors (*Figure 3F*). Collectively, these results indicate that *Mir182/Mir183*-mediated repression of Ago2 is important for mESC self-renewal and proper differentiation.

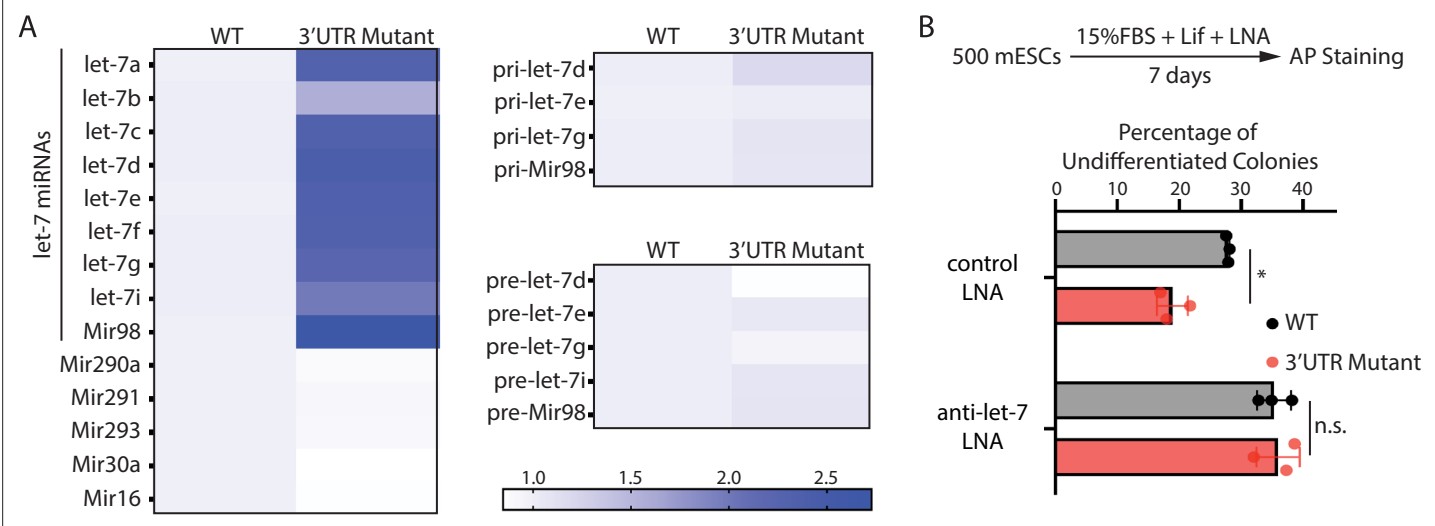

**Figure 4.** The stemness defects in the 3'UTR mutant mouse embryonic stem cells (mESCs) are caused by elevated *let-7* microRNAs (miRNAs). (**A**) Relative levels of miRNAs, *let-7* pri-miRNAs, and *let-7* pre-miRNAs in the wild-type (WT) and the *Ago2* 3'UTR mutant mESCs. For each miRNA, pri-miRNA, and pre-miRNA, the expression level in WT cells was set as one for relative comparison. U6 RNA was used for normalization in miRNA and pre-miRNA quantification; 18 S rRNA was used for normalization in pri-miRNA quantification. The heatmap was generated from the means of three independent replicates. (**B**) Colony formation assay for WT and the *Ago2* 3'UTR mutant mESCs cultured in the presence of 500 nM anti-*let-7* locked nucleic acid (LNA) or a control LNA. The results represent three independent experiments. *p < 0.05, and n.s. not significant (p > 0.05) by the Student's t-test.

The online version of this article includes the following figure supplement(s) for figure 4:

**Source data 1.** Excel files of numbers for *Figure 4A and B*.

## *Mir182/Mir183*-mediated repression of *Ago2* in mESCs inhibits the *let-7* miRNA-mediated differentiation pathway

Two observations lead us to the hypothesis that *Mir182/Mir183*-mediated repression of *Ago2* in mESCs counteracts the differentiation pathway controlled by the *let-7* miRNAs, a conserved miRNA family that promotes stem cell differentiation (*Roush and Slack, 2008*). First, in *Dgcr8Δ* mESCs, where endogenous miRNAs' biogenesis is blocked, ectopic expression of *Mir183* inhibits the stem cell differentiation triggered by exogenous let-7 miRNA (*Wang et al., 2017*). Second, our recent study indicated that increasing Ago2 levels in mESCs results in stemness defects in a *let-7*-miRNA-dependent manner. This specificity on *let-7* miRNAs is because the pro-differentiation *let-7* miRNAs are actively transcribed in mESCs, and the increased Ago2 binds and stabilizes the *let-7* miRNAs that are otherwise degraded in mESCs, thereby promoting mESCs differentiation (*Liu et al., 2021*).

To test this hypothesis, we examined the expression of *let-7* miRNAs. The 3'UTR mutant mESCs had significantly higher *let-7* miRNAs than the WT mESCs (*Figure 4A*). This increase is specific to *let-7* miRNAs because non-let-7 miRNAs were not elevated (*Figure 4A*). Moreover, consistent with our previous observation that increased Ago2 stabilizes mature *let-7* miRNAs (*Liu et al., 2021*), the *pri-let-7* miRNAs and the *pre-let-7* miRNAs were not significantly increased in the 3'UTR mutant mESCs (*Figure 4A*). To determine whether the increased *let-7* miRNAs are responsible for the stemness defects in the 3'UTR mutant mESCs, we inhibited *let-7* miRNAs using locked nucleic acid (LNA) antisense oligonucleotides targeting the conserved seed sequence of *let-7* miRNAs. When *let-7* miRNAs were inhibited, the stemness defects of the 3'UTR mutant mESCs were abolished (*Figure 4B*), indicating that disruption of *Mir182/Mir183*-mediated repression of *Ago2* in mESCs activates differentiation through the *let-7* miRNA pathway.

## *Mir182/Mir183* and trim71 function in parallel to repress *Ago2* mRNA in mESCs

Our previous study indicated that *Ago2* mRNA is also repressed by Trim71 in mESCs (*Liu et al., 2021*). Interestingly, the Trim71-binding site in the 3'UTR of *Ago2* mRNA is different from the

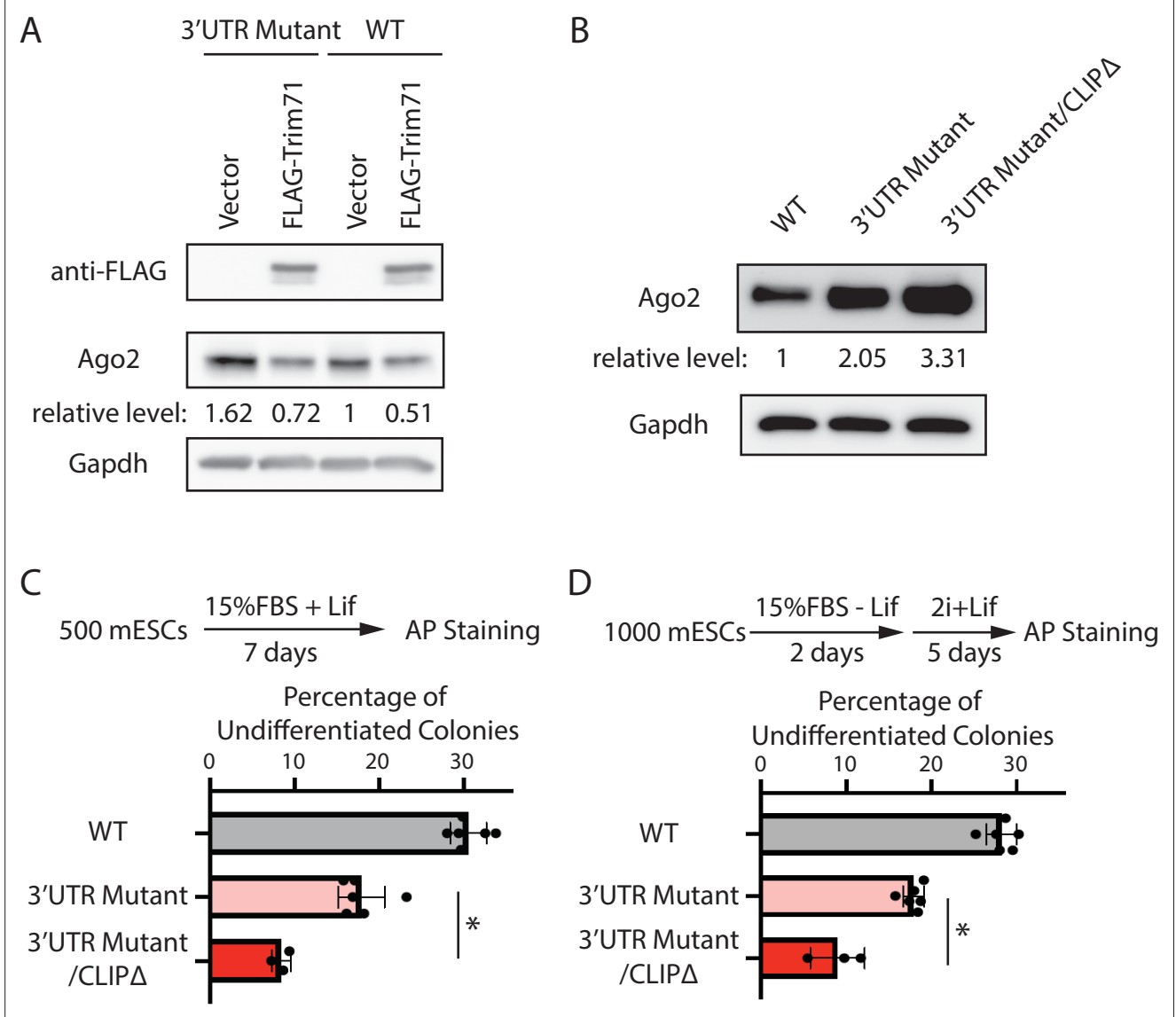

**Figure 5.** *Mir182/Mir183* and Trim71 function in parallel to repress *Ago2* mRNA in mouse embryonic stem cells (mESCs). (**A**) Western blotting in the wild-type (WT) mESCs expressing either a vector or FLAG-Trim71 and in the 3'UTR mutant mESCs expressing either a vector or FLAG-Trim71. (**B**) Western blotting in the WT, 3'UTR mutant, and 3'UTR mutant/CLIPΔ mESCs. In (A and B), GAPDH was used for normalization in calculating the relative expression levels. (**C**) Colony formation assay for mESCs. (**D**) Exit pluripotency assay for mESCs.

The online version of this article includes the following figure supplement(s) for figure 5:

**Source data 1.** Tiff files of raw gel images for *Figure 5A and B*; *Figure 5—figure supplement 1*.

**Figure supplement 1.** Generation of the CLIPΔ in the 3'UTR mutant mouse embryonic stem cells (mESCs).

*Mir182/Mir183*-binding sites, suggesting that *Mir182/Mir183* and Trim71 function in parallel to repress *Ago2* mRNA in mESCs. We performed the following experiments to test this.

At the molecular level, we observed that over-expression of Trim71 still repressed Ago2 in the 3'UTR mutant mESCs (*Figure 5A*), where *Mir182/Mir183*-mediated repression is abolished (*Figure 3*). Moreover, in the 3'UTR mutant mESCs, inhibiting Trim71-mediated repression of Ago2 through deleting the Trim71-binding site in the 3'UTR of *Ago2* mRNA (CLIPΔ) (*Liu et al., 2021*) further increased Ago2 level (*Figure 5B*, *Figure 5—figure supplement 1*). These results indicate that Trim71 and *Mir182/Mir183* independently repress *Ago2* mRNA in mESCs.

At the cell function level, we found that introducing the CLIPΔ in the 3'UTR mutant mESCs further decreased stem cell self-renewal, as determined by the colony formation assay (*Figure 5C*), and

accelerated differentiation, as measured by the exit pluripotency assay (*Figure 5D*). These observations argue that Trim71 and *Mir182/Mir183* function independently in regulating stemness in mESCs through modulating *Ago2* mRNA.

Collectively, these findings indicate that *Mir182/Mir183* and Trim71 function in parallel to repress *Ago2* mRNA in mESCs.

## Discussion

Our data reveal that the predominant Ago protein in mESCs, Ago2, is developmentally regulated, with gradually increasing levels when mESCs exit pluripotency. Two miRNAs abundantly expressed in mESCs, *Mir182/Mir183*, contribute to the repression of *Ago2* in the pluripotent state. This miRNA-mediated regulation of *Ago2* is critical to maintaining stemness. Our findings raise several interesting aspects of miRNAs in stem cell biology.

First, since Ago2 is the predominant Ago protein in mESCs, the Ago2 expression pattern during mESCs' transition from self-renewal to differentiation argues that although certain individual miRNAs may be required for pluripotency (e.g., *Mir182/Mir183*), the global miRNA activity is suppressed in the pluripotent state and induced when mESCs initiate differentiation. Consistent with this notion, knocking out key components in global miRNA biogenesis, such as Dgcr8 (*Wang et al., 2007*), Dicer (*Kanellopoulou et al., 2005*; *Murchison et al., 2005*), or Ago2 in the miRISC (*Liu et al., 2021*), does not negatively affect mESCs self-renewal. However, differentiation in all these mutant mESCs is severely compromised. Thus, at the global level, miRNAs may play more important roles in mESC differentiation.

Second, previous studies indicate that the two components of the miRISC, the Ago protein and its associated miRNA, mutually regulate each other. In the absence of miRNAs, the Ago protein is destabilized (*Martinez and Gregory, 2013*; *Smibert et al., 2013*), while miRNAs are also unstable if they are not associated with Ago proteins (*Winter and Diederichs, 2011*). Thus, the effective miRNA activity depends on the limiting component in the miRISC. Our previous studies indicated that the conserved pro-differentiation *let-7* miRNAs are sensitive to Ago2 levels because an increase of Ago2 results in specific stabilization of *let-7* miRNAs that are otherwise degraded (*Liu et al., 2021*). Thus, for *let-7* miRISC, Ago2 is possibly the limiting component in mESCs. Repression of *Ago2* by either *Mir182/Mir183* as we characterized here or Trim71 as we identified previously (*Liu et al., 2021*) likely limits the effective *let-7* miRISCs. Interestingly, the pro-differentiation *let-7* miRICSs can positively auto-regulate their own biogenesis through inhibiting Lin28a, a conserved *let-7* target, because Lin28a inhibits the biogenesis of *let-7* miRNAs through promoting their pre-miRNA degradation (*Tsialikas and Romer-Seibert, 2015*). Thus, the effective *let-7* miRNAs need to be tightly controlled in stem cells. The two repression mechanisms on *Ago2* mRNA contribute to limiting the amount of effective *let-7* miRISCs and maintaining pluripotency in mESCs. We speculate that similar mechanisms of regulating miRISCs by RNA-binding proteins and miRNAs may exist in other developmental processes. Moreover, Ago2 is dysregulated under many pathological conditions, such as cancer (*Adams et al., 2014*). Thus, regulating miRISCs through modulating Ago2 levels may also contribute to pathogenesis.

Finally, it is noticed that the *Mir182Δ/Mir183Δ* mESCs displayed stronger defects in self-renewal and differentiation than the 3'UTR mutant mESCs did (*Figure 2C and D* versus *Figure 3D and E*). Thus, besides *Ago2* mRNA, *Mir182/Mir183* may regulate additional mRNAs that are important for stem cell biology.

## Materials and methods

### Key resources table

| Reagent type (species) or resource | Designation | Source or reference | Identifiers | Additional information |
|---|---|---|---|---|
| Antibody | (Mouse monoclonal) anti-FLAG M2 | Sigma-Aldrich | Cat# F1804 | WB (1:5000) |
| Antibody | (Mouse monoclonal) anti-GAPDH (6 C5) | Santa Cruz Biotechnology | Cat# sc-32233 | WB (1:5000) |
| Antibody | (Rabbit monoclonal) anti-beta-Tubulin | Selleckchem | Cat# A5032 | WB (1:5000) |
| Antibody | (Rabbit monoclonal) anti-Ago1 (D84G10) | Cell Signaling Technology | Cat# 5053 | WB (1:1000) |

*Continued on next page*

*Continued*

| Reagent type (species) or resource | Designation | Source or reference | Identifiers | Additional information |
|---|---|---|---|---|
| Antibody | (Rabbit monoclonal) anti-Ago2 | Bimake | Cat# A5701 | WB (1:3000) |
| Antibody | (Mouse monoclonal) anti-Oct-4 | BD Transduction Laboratories | Cat# 611202 | WB (1:5000) |
| Antibody | (Rabbit monoclonal) anti-Nanog (D2A3) | Cell Signaling Technology | Cat# 8822 | WB (1:3000) |
| Antibody | Goat Anti-Rabbit IgG (H L)-HRP Conjugate | Bio-Rad | Cat# 170-6515 | WB (1:5000) |
| Antibody | Goat Anti-Mouse IgG (H L)-HRP Conjugate | Bio-Rad | Cat# 170-6516 | WB (1:5000) |
| Chemical compound, drug | DMEM/F-12 | Gibco | Cat# 12500096 | |
| Chemical compound, drug | FBS | Millipore | Cat# ES-009-B | |
| Chemical compound, drug | mLIF | Millipore | Cat# ESG1107 | |
| Chemical compound, drug | PD0325901 | APExBio | Cat# A3013 | |
| Chemical compound, drug | CHIR99021 | APExBio | Cat# A3011 | |
| Chemical compound, drug | N2 | Millipore | Cat# SCM012 | |
| Chemical compound, drug | N27 | Millipore | Cat# SCM013 | |
| Chemical compound, drug | MEM NEAA | Gibco | Cat# 11140–50 | |
| Chemical compound, drug | Penicillin-Streptomycin | Gibco | Cat# 11548876 | |
| Chemical compound, drug | L-Glutamin | Sigma-Aldrich | Cat# G7513 | |
| Chemical compound, drug | β-Mercaptoethanol | Sigma-Aldrich | Cat# M3148 | |
| Chemical compound, drug | Accutase | Millipore | Cat# SF006 | |
| Chemical compound, drug | Fugene6 | Promega | Cat# E2691 | |
| Chemical compound, drug | Puromycin | Sigma-Aldrich | Cat# P9620 | |
| Chemical compound, drug | Doxycycline | Sigma-Aldrich | Cat# D9891 | |
| Chemical compound, drug | Protease inhibitors | Bimake | Cat# B14001 | |
| Chemical compound, drug | Gelatin | Sigma-Aldrich | Cat# G1890 | |
| Chemical compound, drug | One Step-RNA Reagent | Bio Basic | Cat# BS410A | |
| Chemical compound, drug | DNaseI | NEB | Cat# M0303L | |
| Chemical compound, drug | SuperScript II Reverse Transcriptase | Invitrogen | Cat# 18064014 | |
| Chemical compound, drug | SsoAdvanced Universal SYBR Green Supermix | Bio-Rad | Cat# 1725270 | |
| Chemical compound, drug | Q5 High-Fidelity DNA Polymerase | NEB | Cat# M0491L | |
| Chemical compound, drug | Control LNA | Qiagen | Cat# 339137 | |
| Chemical compound, drug | anti-let-7 LNA | Qiagen | Cat# YFI0450006 | |
| Commercial assay or kit | Alkaline Phosphatase Assay Kit | System Biosciences | Cat# AP100R-1 | |
| Commercial assay or kit | Gibson Assembly Master Mix | NEB | Cat# E2611L | |
| Commercial assay or kit | Pierce BCA Protein Assay Kit | Thermo Fisher Scientific | Cat# 23225 | |
| Commercial assay or kit | Mir-X miRNA First Strand Synthesis Kit | Takara | Cat# 638313 | |
| Cell line (*Mus musculus*) | ES-E14TG2a mESC | ATCC | CRL-1821 | |
| Cell line (*Mus musculus*) | FLAG-Ago1 mESC | This paper | | |
| Cell line (*Mus musculus*) | FLAG-Ago2 mESC | PMID:33599613 | | |
| Cell line (*Mus musculus*) | *Mir182Δ* mESC | This paper | | |
| Cell line (*Mus musculus*) | *Mir183Δ* mESC | This paper | | |
| Cell line (*Mus musculus*) | *Mir182Δ/Mir183Δ* mESC | This paper | | |
| Cell line (*Mus musculus*) | 3'UTR Mutant mESC | This paper | | |

*Continued*

| Reagent type (species) or resource | Designation | Source or reference | Identifiers | Additional information |
|---|---|---|---|---|
| Cell line (*Mus musculus*) | *Mir182Δ/Mir183Δ*/3'UTR Mutant mESC | This paper | | |
| Recombinant DNA reagent | PiggyBac-based dox-inducible expression vector | PMID:33599613 | pWH406 | |
| Recombinant DNA reagent | Inducible GFP expressing vector | PMID:33599613 | pWH1055 | |
| Recombinant DNA reagent | Inducible mouse *Mir182* expressing vector | This paper | pWH1039 | |
| Recombinant DNA reagent | Inducible mouse *Mir183* expressing vector | This paper | pWH1040 | |
| Recombinant DNA reagent | sgRNA and Cas9 expressing vector (pX458) pWH464 | Addgene | Cat# 48138 | |
| Recombinant DNA reagent | Super PiggyBac Transposase expressing vector (pWH252) | System Biosciences | Cat# PB210PA-1 | |

All the antibodies, plasmids, and oligonucleotides used in this study are listed in *Supplementary file 1*.

## Cell lines

All the cell lines from this study are based on ES-E14TG2a mESC (ATCC, CRL-1821). They are listed in *Supplementary file 1*. The ES-E14TG2a mESCs were authenticated through STR profiling and were negative for mycoplasma contamination determined by a PCR-based kit.

## CRISPR/Cas9-mediated genome editing in mESCs

To generate the FLAG-Ago1, FLAG-Ago2 mESCs, or *Ago2* 3'UTR mutant mESCs, cells were co-transfected with 2 µg of pWH464 (pSpCas9(BB)-2A-GFP [pX458]) expressing the corresponding targeting sgRNA and 1 µg of the corresponding donor oligo or plasmid using the Fugene6 (Promega). To generate *Mir182Δ* and *Mir183Δ* mESCs, cells were transfected with 2 µg of pWH464 expressing a pair of sgRNAs targeting *pri-Mir182* or *pri-Mir183*. The transfected cells were subject to single cell sorting and the resulting clones were subject to genotyping to identify the correct clones.

## qRT-PCR

For pri-miRNA quantification, reverse transcription was performed using random hexamers and Superscript II Reverse Transcriptase. Pre-miRNA and miRNA quantifications were using the Takara's Mir-X miRNA quantification method. qPCR was performed in triplicate for each sample using the SsoAdvanced Universal SYBR Green Supermix (Bio-Rad) on a CFX96 real-time PCR detection system (Bio-Rad).

## Western blotting

Proteins were harvested in RIPA buffer (10 mM Tris-HCl pH 8.0, 140 mM NaCl, 1 mM EDTA, 0.5 mM EGTA, 1 % Triton X-100, 0.1 % sodium deoxycholate, 0.1 % SDS, and protease inhibitor cocktail) and quantified with a BCA Protein Assay Kit (Thermo Fisher Scientific). Equal amounts of protein samples were resolved by SDS-PAGE, and then transferred to PVDF membranes. Western blotting was performed using a BlotCycler (Precision Biosystems) with the corresponding primary and secondary antibodies. The membranes were then treated with the Western ECL substrate (Bio-Rad), and the resulting signal was detected using an ImageQuant LAS 500 instrument (GE Healthcare).

## Colony formation assay and exit pluripotency assay

For colony formation assay, 500 cells were plated on a 12-well plate in 2i + Lif media or Lif media (DMEM/F12 supplemented with 15 % FBS, 1× penicillin/streptomycin, 0.1 mM non-essential amino acids, 2 mM L-glutamine, 0.1 mM 2-mercaptoethanol, and 1000 U/ml Lif). For exit from pluripotency assay, 1000 cells were plated on a gelatin-coated six-well plate in differentiation media (DMEM/F12 supplemented with 15 % FBS, 1× penicillin/streptomycin, 0.1 mM non-essential amino acids, 2 mM L-glutamine, and 0.1 mM 2-mercaptoethanol) for 2 days, then cultured in 2i + Lif media for another 5 days. Colonies were stained using AP staining kit and grouped by differentiation status 6–7 days after plating.

## Embryoid body formation

For differentiation via embryoid body (EB) formation, $3 \times 10^6$ cells were plated per 10 cm bacterial grade Petri dish and maintained on a horizontal rotator with a rotating speed of 30 rpm in differentiation media. The resultant EBs were harvested at the indicated time points.

RNA antisense purification mESCs were crosslinked with 0.1 % formaldehyde for 5 min at room temperature, and the crosslinking reaction was quenched by adding 1/20 volume of 2.5 M glycine and incubating the mESCs at room temperature for 10 min on a rotating platform. The cells were then harvested and lysed in cell lysis buffer (50 mM Tris-HCl pH 7.4, 150 mM NaCl, 5 mM EDTA, 10 % glycerol, 1 % Tween-20, with freshly added proteinase inhibitors). The cell lysate was cleared by centrifugation at 20,000 $g$ for 10 min at 4 °C. The resulting supernatant was used for RNA antisense purification; 5 mg lysate in 500 µl lysis buffer was used for each purification. Specifically, a set of 5'-end biotinylated anti-sense DNA oligos and 5 µl RNase inhibitor (NEB) were added to the lysate, resulting in a final concentration of 0.1 µM for each oligo. The lysate was incubated at room temperature for 1 hr on a rotating platform. Then, 100 µl Dynabeads MyOne Streptavidin C1 (Invitrogen) was added and the lysate further incubated for 30 min at room temperature on a rotating platform. The magnetic beads were isolated through a magnetic stand and then subject to four washes, with each wash in 500 µl high salt wash buffer (5× PBS, 0.5 % sodium deoxycholate, 1 % Triton X-100). The washed beads were resuspended in 100 µl DNaseI digestion mix (1 × DNaseI digestion buffer with 5 µl DNaseI [NEB]) and incubated at 37 °C for 20 min, followed by adding 350 µl LET-SDS buffer (25 mM Tris-HCl pH 8.0, 100 mM LiCl, 20 mM EDTA pH 8.0, 1% SDS) and 50 µl proteinase K (20 mg/ml, Thermo Fisher Scientific). The beads were then incubated on a thermomixer at 55 °C 1000 rpm for 2 hr. The RNA was isolated through phenol extraction and isopropanol precipitation with glycoblue (Ambion) as a coprecipitant.

## Acknowledgements

We thank Dr Xiaoli Chen for his assistance with miRNA prediction. This work is supported by Mayo Foundation for Medical Education and Research.

## Additional information

### Funding

| Funder | Grant reference number | Author |
|---|---|---|
| Mayo Foundation for Medical Education and Research | | Qiuying Liu<br>Mariah K Novak<br>Rachel M Pepin<br>Taylor Eich<br>Wenqian Hu |

The funders had no role in study design, data collection and interpretation, or the decision to submit the work for publication.

### Author contributions

Qiuying Liu, Data curation, Formal analysis, Investigation, Methodology, Writing - original draft, Writing - review and editing; Mariah K Novak, Rachel M Pepin, Taylor Eich, Data curation, Investigation, Methodology, Writing - review and editing; Wenqian Hu, Conceptualization, Data curation, Funding acquisition, Investigation, Methodology, Project administration, Supervision, Writing - original draft, Writing - review and editing

### Author ORCIDs

Qiuying Liu http://orcid.org/0000-0002-1474-4487
Wenqian Hu http://orcid.org/0000-0003-3577-3604

### Decision letter and Author response

Decision letter https://doi.org/10.7554/eLife.72289.sa1
Author response https://doi.org/10.7554/eLife.72289.sa2

## Additional files

### Supplementary files
• Supplementary file 1. Antibodies, plasmids, and oligonucleotides used in this study.
• Transparent reporting form

### Data availability
All data generated or analyzed during this study are included in the manuscript and supporting files. Source data files have been provided for Figures 1–5.

The following previously published datasets were used:

| Author(s) | Year | Dataset title | Dataset URL | Database and Identifier |
|---|---|---|---|---|
| Marks H, Menafra R, Kalkan T, Denissov S, Jones K, Hofemeister H, Nichols J, Kranz A, Stewart AF, Smith A, Stunnenberg HG | 2012 | Epigenome and transcriptome of naive pluripotent mouse embryonic stem (ES) cells cultured in 2i serum-free medium | https://www.ncbi.nlm.nih.gov/geo/query/acc.cgi?acc=GSE23943 | NCBI Gene Expression Omnibus, GSE23943 |
| Hu W, Liu Q, Zhang H, Chen X, Zhang S | 2021 | Studies on Trim71 in mouse embryonic stem cells | https://www.ncbi.nlm.nih.gov/geo/query/acc.cgi?acc=GSE138284 | NCBI Gene Expression Omnibus, GSE138284 |

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
