## [Decision Letter]

**Acceptance summary:**

In aggregate, the data strongly supports the conclusions made. This revised manuscript includes additional data and clarifications: the authors have responded effectively to comments raised in initial review.

**Decision letter after peer review:**

[Editors’ note: the authors submitted for reconsideration following the decision after peer review. What follows is the decision letter after the first round of review.]

Thank you for submitting the paper "microRNA-mediated regulation of microRNA machinery controls cell fate decisions" for consideration by *eLife*. Your article has been reviewed by 2 peer reviewers, and the evaluation has been overseen by a Reviewing Editor and a Senior Editor.

We are sorry to say that, after consultation with the reviewers, we have decided that this work cannot be considered further for publication by *eLife*.

The agreement was that the submission was not suitable as a Research Advance. One reviewer judged the results to be a relatively small advance over your earlier paper. The other reviewer commented that a significant number of experiments were required before this could be considered for publication in *eLife*. Please see the detailed comments of the reviewers below.

*Reviewer #1:*

The study by Liu and colleagues investigated the molecular mechanisms that regulate the function of Argonaute 2 (AGO2), an essential component of the RNA-induced silencing complex, to control the developmental progression of pluripotent mouse embryonic stem cells (mESCs). Through a series of in vitro molecular assays, the authors showed that AGO2 is the major developmentally regulated Argonaute protein in mESCs, and that AGO2 is repressed by microRNA-182/microRNA-183. The experiments appear well-designed and the methods are technically sound. However, the study provides only marginal conceptual advances, especially given the very recent publication of a study by the same group showing how AGO2 is regulated by the heterochronic gene Trim71 (Liu et al., 2021, *eLife*). I have two major concerns for the authors to consider:

First, is there a way for the authors to confirm direct binding of microRNA-182/microRNA-183 to Ago2 RNA? If so, I consider this an essential experiment for the authors to carry out, given that the evidence thus far appears indirect (in sillico analysis showing presence of binding motifs and altered expression of Ago2 when the putative binding sites were mutated in cell lines).

Second, the authors have already shown in a previous work that Trim71 also represses Ago2. It is thus surprising to me that in the current study, the authors make no attempt to experimentally link how microRNA-182/microRNA-183 may be working together with Trim71 to regulate Ago2. What is the significance to the overall field to show identify and show additional repressors of Ago2, and how do these newly identified repressors cooperate with Trim71? It is essential for the authors to better link their current work in the context of previous findings by demonstrating how Ago2 repressors may be working together, and whether they may specific functions that distinguish them from each other.

From an experimental standpoint, the authors need to provide better mechanistic insights that explain how microRNA-182/microRNA-183 may cooperate with Trim71 to regulate Ago2 function in mESCs. Otherwise, the conceptual advance provided by the current manuscript appears minor.

*Reviewer #2:*

Liu et al. investigated the role of micorRNA(miRNA)-mediated regulation of miRNA-induced silencing complex (miRISC) during the differentiation of mouse embryonic stem cells (mESCs). Corroborating previous evidences establishing high mRNA levels for only one paralog of Argonautes (Ago2) in mESCs, the authors demonstrated that the Ago2 is expressed at high levels in mESCs at protein levels too, and that the AGO2 levels are elevated upon mESC differentiation. Using many independent and orthogonal experiments, the authors find with abundance of evidence that miRNAs, miR-182 and miR-183 whose conserved bindings are present in the 3'UTR of Ago2 and whose expression negatively correlates with AGO2 levels in mESCs directly repress AGO2 levels and that the loss of this miRNA-mediated repression of AGO2 promotes enhanced differentiation of mESCs in vitro. The strongest evidence of this direct regulation of AGO2 comes from the experiments where miR-182/183 sites are deleted in the AGO2 3'UTR (3'UTR mutant), which recapitulates the phenotype. While these results robustly show that AGO2/miR-182/183 axis enhances mESC differentiation during the induced transition of mESCs from the ground state to the primed state, this work doesn't show whether this axis is sufficient to control the differentiation in the ground/naive state. Finally, in concordance with their prior findings (Liu et al. 2021), the authors show that the AGO2 levels modulated by miR-182/183 also control the let-7-dependent differentiation of mESCs.

While the previous reports showed that miR-182/183 suppresses differentiation-inducing miRNAs, including let-7, miR-26 and miR-218, this work forms a unique niche and provides a link between miR-182/183 and let-7 in the form of miR-182/183-mediated repression of AGO2. The experiments in this paper follow a logical progression and provide high quality data. The conclusions are mostly supported by the data.

However, in its current form, this paper needs clarification on some fundamental aspects and some additional data. See below for the details.

1) The authors use as readout the reduction in colony formation during the transition of mESCs to the primed state, where the undifferentiated colonies are already reduced to 30%, and where there could be many confounding effects from large transcriptional changes. It is surprising that the authors didn't use the colony formation in naive/ground state as the readout and show that the increased AGO2 levels cause cells to lose stemness. They could potentially see much larger effects in the ground state, attributable directly to AGO2/miR-182/183. If using naive/ground state in these experiments is experimentally not feasible, the authors should discuss that.

2) Though the authors conclude that the elevated levels of AGO2 selectively stabilizes let-7 family members, the mechanism of this selectively is lacking. This model might be an over-simplification, since the increased AGO2 levels can cause system-wide cascade of direct and indirect effects, some of which might be responsible for this increased selectivity. It is likely that LIN28A, which is expressed at higher levels in mESCs and is involved in the well-established feedback loop with let-7 miRNA family, is involved in this let-7-specificity. A plausible alternative hypothesis is that the increased levels of AGO2 in 3'UTR mutant or during differentiation stabilizes all actively produced mature miRNAs equally. However, most miRNAs may not cause an immediate impact on levels of other miRNAs or the miRNA machinery, but the newly formed let-7-miRISCs reduce the high levels of LIN28A, promoting the elevated levels of mature let-7 miRNAs, which competes with other mature miRNAs and saturates the newly formed AGO2 molecules. This hypothesis doesn't assume that the increased AGO2 selectively stabilizes let-7 solitarily.

3) The authors claim that AGO2 repression via miR-182/183 is important for proper differentiation of mESC, however, the data in this paper doesn't show an impact of AGO2 repression on defective differentiation.

Some potential outstanding questions that the authors may want to explore towards this paper or the future work are: (1) how do miR-182/183 levels are modulated during the differentiation process and if modulating their levels is sufficient to promote differentiation in the ground/naive state; (2) this paper suggests that the inhibition of many differentiation-inducing miRNAs that has been shown previously to be trigged by miR-182/183 (Wang et al., 2017), is orchestrated through miR-182/183 mediated suppression of miRISC machinery. It would be of great importance to investigate how do miR-182/183 themselves escape the effect of inhibited miRISC machinery, e.g., do these miRNAs have higher half-life in mESCs? (3) Does the loss of AGO2 repression only enhance the differentiation or does it also trigger defective differentiation? (4) What is the significance of this regulation in vivo.

"Two lines of evidence indicated that miR-182/miR-183 targets Ago2 mRNA." – should use "regulates" instead of "targets", since the evidence for direct targeting comes later in the paper.

[Editors’ note: further revisions were suggested prior to acceptance, as described below.]

Thank you for submitting your article "microRNA-mediated regulation of microRNA machinery controls cell fate decisions" for consideration by *eLife*. Your article has been reviewed by 3 peer reviewers, and the evaluation has been overseen by a Timothy Nilsen as the Reviewing Editor and James Manley as the Senior Editor. The following individuals involved in review of your submission have agreed to reveal their identity: Ravi K Patel (Reviewer #1).

Essential revisions:

The reviewers and the reviewing editor found the work to be an interesting extension of your previous findings. Nevertheless, all of the reviewers have made relatively minor suggestions for improvement. In particular, as noted by reviewers 1 and 3, it is important to temper your conclusion regarding the role of let-7. Please address these issues as thoroughly as possible before resubmitting.

*Reviewer #1:*

In this research advance article, Liu et al. investigated the role of micorRNA(miRNA)-mediated regulation of miRNA-induced silencing complex (miRISC) during the differentiation of mouse embryonic stem cells (mESCs). The authors demonstrated that the Ago2 is expressed at high levels in mESCs at protein levels and that the AGO2 levels are elevated upon mESC differentiation. Using multiple independent and orthogonal experiments, the authors find that miRNAs, miR-182 and miR-183 whose conserved binding sites are present in the 3'UTR of Ago2 and whose expression negatively correlates with AGO2 levels in mESCs directly repress AGO2 levels and that the loss of this miRNA-mediated repression of AGO2 promotes enhanced differentiation of mESCs in vitro. The strongest evidence of this direct regulation of AGO2 comes from the experiments where miR-182/183 sites are deleted in the AGO2 3'UTR (3'UTR mutant), which recapitulates the phenotype. These results robustly show that AGO2/miR-182/183 axis enhances mESC differentiation. Finally, in concordance with their prior findings (Liu et al. 2021), the authors show that the AGO2 levels modulated by miR-182/183 or Trim71 control the let-7-dependent differentiation of mESCs. The experiments in this manuscript follow a logical progression and provide high quality data. The conclusions are supported by the data.

*Reviewer #2:*

In aggregate, the data supports strongly the conclusions made. This revised manuscript includes additional data and clarifications: the authors have responded effectively to comments raised in review.

*Reviewer #3:*

In this manuscript Liu and colleagues follow-up on their previous work showing that Ago2 expression is regulated in mESC by Trim-71 by proposing that Ago2 expression is also directly regulated by miR-182/miR-183 and that loss of this interaction affect mESC differentiation by resulting in let-7 stabilization.

Using elegant genetic and biochemical approaches they convincingly show that Ago2 is directly repressed by miR-183/miR-183, as either loss of these two miRNAs or mutation of their binding sites in the Ago2 3'UTR results in upregulation of Ago2 expression by approximately 2 fold.

They also show that targeted inactivation of miR-182/183 and, to a slightly lesser extent, mutation of miR-182/miR-183 binding sites in Ago2's 3'UTR promote differentiation of mESC in vitro.

The manuscript is very well written, the experiments are elegant and include all the appropriate controls, and their results are largely consistent with the model proposed by the authors. While the authors convincingly demonstrate that miR-182/182 directly regulate Ago2, it is slightly less clear whether the proposed increased activity of let-7 can fully explain the observed phenotype.

Overall this is a solid manuscript, although as discussed below not all conclusions are supported by conclusive evidence. I concur with reviewer 1 that the study provides relatively minor conceptual advance, but it can be argued that these results will be of interest to specialists in the field of miRNa and ESC biology.

A major concern I have regards the strength of the conclusion that the impaired stem cell function upon loss of miR-182/183 activity on Ago2 is mediated by increased activity of let-7. The LNA-based experiments are suggestive, but although they prove that blocking let-7 helps preventing differentiation, they do not prove that increased let-7 function underlies the observed phenotype in the Ago2 UTR mutant. In fact, the only evidence that let-7 activity is increased is a modest upregulation of let-7 members expression. Whether this is sufficient to affect gene expression to a detectable extent is unclear. The evidence would be strengthened substantially if the authors showed increased let-7-mediated gene repression in the UTR mutant. A straightforward and relatively inexpensive experiment would be to perform RNAseq analysis on the wt and UTR-mutant mESCs. The authors' model leads to the prediction that the a relatively selective de-repression of predicted let-7 targets should be observed in the UTR mutant samples.

1) Figures 2C and 2D appear swapped in their description in the main text (lines 107-114)

2) lines 132-135: I am not sure the authors can define the effect of the 3'UTR as not "in cis". Even if mediated by loss of miRNA binding, as is likely, the effect would still be on the transcript harboring the mutation, so technically, as far as I know, in cis. Perhaps a different language would be more appropriate. For example, the authors could say that the fact that deletion of miR-182/183 in a the 3'UTR-mut background doesn't result in additional upregulation of Ago2 provides additional evidence that the UTR mutation results in Ago2 upregulation by preventing miR-182/183 binding.

3) It appears to me that loss of miR-182/183 has a stronger effect on ESC differentiation than mutation of their Ago2 3'UTR binding sites. This suggests that miR-182 promote ESC pluripotency by additional mechanism independent from Ago2. Perhaps the authors should comment on this in the discussion.

---

## [Author Response]

[Editors’ note: the authors resubmitted a revised version of the paper for consideration. What follows is the authors’ response to the first round of review.]

Reviewer #1:The study by Liu and colleagues investigated the molecular mechanisms that regulate the function of Argonaute 2 (AGO2), an essential component of the RNA-induced silencing complex, to control the developmental progression of pluripotent mouse embryonic stem cells (mESCs). Through a series of in vitro molecular assays, the authors showed that AGO2 is the major developmentally regulated Argonaute protein in mESCs, and that AGO2 is repressed by microRNA-182/microRNA-183. The experiments appear well-designed and the methods are technically sound. However, the study provides only marginal conceptual advances, especially given the very recent publication of a study by the same group showing how AGO2 is regulated by the heterochronic gene Trim71 (Liu et al., 2021, eLife). I have two major concerns for the authors to consider:

We appreciate that this reviewer believes that “the experiments appear well-designed and the methods are technically sound”. We provided our response to his/her specific comments below.

First, is there a way for the authors to confirm direct binding of microRNA-182/microRNA-183 to Ago2 RNA? If so, I consider this an essential experiment for the authors to carry out, given that the evidence thus far appears indirect (in sillico analysis showing presence of binding motifs and altered expression of Ago2 when the putative binding sites were mutated in cell lines).

We agree with the reviewer that showing miR-182/miR-183 associate with *Ago2* mRNA will strengthen the conclusions of the manuscript.

To address this point, we performed a modified RNA anti-sense purification (RAP) approach (Figure 1—figure supplement 2A). Specifically, to trap potential transient and dynamic RNA-mediated interactions (e.g., microRNA-mediated regulation), we crosslinked the mESCs with 0.1% formaldehyde and then purified target mRNAs from the cell lysate using a set of biotinylated antisense DNA oligoes and streptatividin magnetic beads. The following observations from this experiment indicate that miR182/miR-183 specifically associate with Ago2 mRNA in mESCs.

First, the RAP approach can specifically isolate the target mRNAs (Figure 1—figure supplement 2B). Using anti-sense oligoes target *Ago2* mRNAs, we observed *Ago2* mRNAs, but not control mRNAs (e.g., *Gapdh* mRNA, *Vim* mRNAs), were significantly enriched in the RAP sample.

Second, we found that miR-182/miR-183 were specifically enriched in the RAP-Ago2 mRNA sample, but not RAP-Vim mRNA sample, arguing that these two miRNAs are associated with Ago2 mRNAs (Figure 1—figure supplement 2C).

Third, when the RAP was performed in the 3’UTR mutant mESCs, where the miR182/miR-183 binding sites were mutated (Figure 3A), we observed that the enrichment of miR-182/miR-183 in the RAP-Ago2 mRNA was significantly decreased (Figure 1—figure supplement 2C).

Altogether, these results indicate that miR-182/miR-183 specifically associated Ago2 mRNA in mESCs.

In the revised manuscript, we presented these results in Figure 1—figure supplement 2, and described the experimental procedures in the methods section.

Second, the authors have already shown in a previous work that Trim71 also represses Ago2. It is thus surprising to me that in the current study, the authors make no attempt to experimentally link how microRNA-182/microRNA-183 may be working together with Trim71 to regulate Ago2. What is the significance to the overall field to show identify and show additional repressors of Ago2, and how do these newly identified repressors cooperate with Trim71? It is essential for the authors to better link their current work in the context of previous findings by demonstrating how Ago2 repressors may be working together, and whether they may specific functions that distinguish them from each other.

We appreciate the reviewer’s great suggestion on improving our manuscript. We performed additional experiments to address his point.

Specifically, the different binding locations of Trim71 and miR-182/miR-183 in the 3’UTR of *Ago2* mRNA led us to test whether or not the Trim71-mediated repression of Ago2 mRNA and the miR-182/miR-183-mediated repression of Ago2 mRNA functions independently in mESCs. We found that at the Ago2 expression level, in the absence of miR-182/miR-183-mediated repression (in the 3’UTR mutant mESCs), Trim71 still represses Ago2. Moreover, at the stem cell function level, we observed that inhibition of Trim71-mediated repression of *Ago2* mRNA (through deleting the Trim71-binding site in the 3’UTR of *Ago2* mRNA (CLIPD)) further decreased stem cell self-renewal and accelerated differentiation in the 3’UTR mutant mESCs. Collectively, these observations indicate that miR-182/miR-183 and Trim71 function in parallel to repress *Ago2 mRNA* in mESCs.

We presented these new results as Figure 5 and Figure 5 —figure supplement 1 and added a new section (from lane 176 to lane 197) in the revised manuscript.

From an experimental standpoint, the authors need to provide better mechanistic insights that explain how microRNA-182/microRNA-183 may cooperate with Trim71 to regulate Ago2 function in mESCs. Otherwise, the conceptual advance provided by the current manuscript appears minor.

This point is the same as the point 2 of the public review from this reviewer. Please see our response above.

Reviewer #2:Liu et al. investigated the role of micorRNA(miRNA)-mediated regulation of miRNA-induced silencing complex (miRISC) during the differentiation of mouse embryonic stem cells (mESCs). Corroborating previous evidences establishing high mRNA levels for only one paralog of Argonautes (Ago2) in mESCs, the authors demonstrated that the Ago2 is expressed at high levels in mESCs at protein levels too, and that the AGO2 levels are elevated upon mESC differentiation. Using many independent and orthogonal experiments, the authors find with abundance of evidence that miRNAs, miR-182 and miR-183 whose conserved bindings are present in the 3'UTR of Ago2 and whose expression negatively correlates with AGO2 levels in mESCs directly repress AGO2 levels and that the loss of this miRNA-mediated repression of AGO2 promotes enhanced differentiation of mESCs in vitro. The strongest evidence of this direct regulation of AGO2 comes from the experiments where miR-182/183 sites are deleted in the AGO2 3'UTR (3'UTR mutant), which recapitulates the phenotype. While these results robustly show that AGO2/miR-182/183 axis enhances mESC differentiation during the induced transition of mESCs from the ground state to the primed state, this work doesn't show whether this axis is sufficient to control the differentiation in the ground/naive state. Finally, in concordance with their prior findings (Liu et al. 2021), the authors show that the AGO2 levels modulated by miR-182/183 also control the let-7-dependent differentiation of mESCs.While the previous reports showed that miR-182/183 suppresses differentiation-inducing miRNAs, including let-7, miR-26 and miR-218, this work forms a unique niche and provides a link between miR-182/183 and let-7 in the form of miR-182/183-mediated repression of AGO2. The experiments in this paper follow a logical progression and provide high quality data. The conclusions are mostly supported by the data.However, in its current form, this paper needs clarification on some fundamental aspects and some additional data. See below for the details.

We appreciate that this reviewer believes that “this work forms a unique niche” and “The experiments in this paper follow a logical progress and provide high quality data. The conclusions are mostly supported by the data”. We provided our response to his/her specific comments below.

1) The authors use as readout the reduction in colony formation during the transition of mESCs to the primed state, where the undifferentiated colonies are already reduced to 30%, and where there could be many confounding effects from large transcriptional changes. It is surprising that the authors didn't use the colony formation in naive/ground state as the readout and show that the increased AGO2 levels cause cells to lose stemness. They could potentially see much larger effects in the ground state, attributable directly to AGO2/miR-182/183. If using naive/ground state in these experiments is experimentally not feasible, the authors should discuss that.

The naïve/ground state mESCs were maintained through culturing mESCs in the presence of two chemical inhibitors: PD0325901 and CHIR-99021, which block mitogen-activated protein kinase (MEK1) and glycogen synthase kinase-3, respectively. These inhibitions potently block the stem cell differentiation, thereby maintaining mESCs in the naïve/ground state. Although this condition provides high percentage of AP+ colonies in the colony formation assay, the strong inhibition of differentiation by the chemical inhibitors will “mask” many pro-differentiation effects. Thus, the colony formation assay is usually performed in the 15% FBS + Lif medium, where both prostemness and pro-differentiation phenotypes can be observed.

In the revised manuscript, we add this rational so that general readers can understand why we did the stemness assays in the 15% FBS + Lif medium instead of the 2i+lif medium (line 109 of the revised manuscript).

2) Though the authors conclude that the elevated levels of AGO2 selectively stabilizes let-7 family members, the mechanism of this selectively is lacking. This model might be an over-simplification, since the increased AGO2 levels can cause system-wide cascade of direct and indirect effects, some of which might be responsible for this increased selectivity. It is likely that LIN28A, which is expressed at higher levels in mESCs and is involved in the well-established feedback loop with let-7 miRNA family, is involved in this let-7-specificity. A plausible alternative hypothesis is that the increased levels of AGO2 in 3'UTR mutant or during differentiation stabilizes all actively produced mature miRNAs equally. However, most miRNAs may not cause an immediate impact on levels of other miRNAs or the miRNA machinery, but the newly formed let-7-miRISCs reduce the high levels of LIN28A, promoting the elevated levels of mature let-7 miRNAs, which competes with other mature miRNAs and saturates the newly formed AGO2 molecules. This hypothesis doesn't assume that the increased AGO2 selectively stabilizes let-7 solitarily.

We appreciate this reviewer’s very insightful comments on the potential mechanisms by which the increased Ago2 specifically stabilizes and increases let-7 miRNAs in mESCs. Indeed, the data from our previous paper (Liu et al., 2021 *eLife*) indicated that Lin28a plays an important role in the let-7’s specific response to Ago2 levels, which is exactly as the reviewer predicted. Specifically, in Figure 5 of the previous *eLife* paper, we showed that when Ago2 level was induced, in addition to the specific increase of let7 mature miRNAs, there were also specific decrease of both Lin28a and Trim71, the two conserved (from *C. elegans* to human) targets of let-7 microRNAs. Thus, Ago2, let7 miRNAs, and Lin28a forms a delicate regulatory loop, which makes let-7 mature miRNA level sensitive to Ago2 levels. Specifically, a slight increase of the let-7 miRNAs caused by elevated Ago2 decreases Lin28a. This decrease alleviates Lin28a-mediated inhibition on the maturation of let-7 miRNAs, resulting in more let-7 pre-miRNAs become mature let-7 miRNAs, which further decreases Lin28a levels and prevents more let-7 pre-miRNA from Lin28a-mediated degradation. Thus, this positive regulatory loop amplifies let-7 miRNAs and makes the pro-differentiation let-7 miRNAs sensitive to Ago2 levels in stem cells. We made this point in the Discussion section of the previous *eLife* paper.

Although the regulatory loop involving Lin28a is the most likely explanation, we agree with the reviewer that it is difficult to identify the exact causal factor on why let-7 is specifically increased when Ago2 is elevated, using the in vivo system. Because as the reviewer nicely points out “the increased AGO2 levels can cause system-wide cascade of direct and indirect effects, some of which might be responsible for this increased selectivity.” We believe a complete resolution on this issue would require an in vitro system that can recapitulate these regulatory events, which can limit the potential system-wide cascade of indirect effects and enable specific and direct studies on the interplay among let-7 miRNA (pre-miRNA), Lin28a, Ago2, and Dicer (pre-miRNA processing machinery). Unfortunately, however, currently we don’t know such an in vitro system exists. Thus, we acknowledge the limitations in interpreting the results from the in vivo studies in mESCs.

As discussed above, our current work is closely related to the previous *eLife* paper. Thus, we feel the Research Advance format of *eLife*, which can be linked to the previous *eLife* paper, would be an ideal format for us to present these findings.

3) The authors claim that AGO2 repression via miR-182/183 is important for proper differentiation of mESC, however, the data in this paper doesn't show an impact of AGO2 repression on defective differentiation.

We feel the definition on “defective differentiation” mentioned by the reviewer is vague to us. Because “defective differentiation” can suggest many scenarios (e.g., mESCs do not differentiate at all, differentiate faster or slower, etc.). If we define the “defective differentiation” as any differentiation process that is different from the WT mESCs differentiation, then we believe that our results from this manuscript and the previous *eLife* paper shows that repressing Ago2 expression (by either miR-182/miR-183 in this study or Trim71 as described in the previous study) prevents mESCs from defective differentiation (as loss of these regulations accelerate the differentiation process of mESCs).

Some potential outstanding questions that the authors may want to explore towards this paper or the future work are:

We appreciate the reviewer’s thoughtful comments and suggestions on this work and on our future studies. We provide our response below.

1) How do miR-182/183 levels are modulated during the differentiation process and if modulating their levels is sufficient to promote differentiation in the ground/naive state;

First, previous genomic studies and CRISPR/Cas9-mediated functional studies (Pulecio et al., 2017; Rajagopal et al., 2016) indicate that at the transcriptional level, the production of miR-182 and miR-183 is regulated by the pluripotency factor Nanog in mESCs. These observations explain, in part (at the production level), why these miRNAs are highly expressed in mESCs.

Second, the ground state of mESCs is a transient developmental stage, and in culture, this state is normally maintained via the two chemical inhibitors (PD0325901 and CHIR99021). As discussed in the first point of the public review from this reviewer, these two chemical inhibitors “mask” many pro-differentiation effects. The following two observations (from our study and the literature), obtained from standard mESC culture conditions (15% FBS + lif), however, argues that inhibition of miR-182/miR-183 triggers differentiation: a) the miR-182/mi-183 knockout mESCs have very few AP+ colonies compared to the WT mESCs (Figure 2C); b) in Dgcr8 knockout mESCs, where normal differentiation is blocked due to the absence of endogenous miRNAs, miR-182/miR-183 can inhibit the strong differentiation effects induced by let-7 miRNAs (Wang et al., 2017). These results from both loss-of-function and gain-of-function studies strongly argue that decrease of miR-182/miR-183 levels can trigger differentiation in mESCs, and miR-182/miR-183 play important roles in maintaining pluripotency.

2) this paper suggests that the inhibition of many differentiation-inducing miRNAs that has been shown previously to be trigged by miR-182/183 (Wang et al., 2017), is orchestrated through miR-182/183 mediated suppression of miRISC machinery. It would be of great importance to investigate how do miR-182/183 themselves escape the effect of inhibited miRISC machinery, e.g., do these miRNAs have higher half-life in mESCs?

The reviewer raised a great question regarding regulations of miRISC in mESCs. The interpretation of the results could be: a) miR-182/183 have unique features that enable them to escape the effects of repressed miRISC machinery, as the reviewer suggested; or alternatively, b) among miRNAs expressed in mESCs, let-7s are uniquely sensitive to miRISC machinery.

To discriminate these two possibilities, we measured the stabilities of miR-183/miR-183 and let-7 miRNAs in mESCs maintained in the ground state (2i+lif) and differentiating mESCs (-lif medium) (Author response image 1):

**Author response image 1. sa2fig1:** 

These results indicate that there was no change of miR-182/miR-183 stability when mESCs were either in the ground state or in the differentiating state. Interestingly, however, let-7 miRNAs (let-7a, let-7f, miR-98) have significantly higher stability in differentiating mESCs than that in the ground state mESCs. This increase of stability correlates with Ago2 levels, as differentiating mESCs have higher Ago2 level than ground state mESCs have (Figure 1C). Combined with our previous observation that increase of Ago2 specifically increase and stabilize let-7 miRNAs in mESCs (Figure 5 and Figure 5 —figure supplement 2 in Liu et al., 2021 *eLife*), these observations argue that it is the let-7 miRNAs that are uniquely sensitive to Ago2 levels in mESCs.As the reviewer pointed out in the point 2) of the public review, and as we discussed in the previous *eLife* paper (Liu et al., 2021 *eLife*), this let-7 miRNAs’ unique sensitive to Ago2 levels could be due to the deliciated positive autoregulatory loops among let-7 miRNAs, Ago2, and Lin28a (a conserved let-7 target).

3) Does the loss of AGO2 repression only enhance the differentiation or does it also trigger defective differentiation?

As discussed in the point 3) of the public review, we feel that the meaning of “defective differentiation” is not clear to us. If “defective differentiation” is defined as any differentiation process that is different from the differentiation of WT mESCs, then loss of Ago2 repression results in accelerated mESC differentiation, which is a form of defective differentiation.

4) What is the significance of this regulation in vivo.

We completely agree with the reviewer that dissecting the in vivo relevance of this regulation (miR-182/miR-183 mediated regulation of Ago2) using animal models is of high significance for future work. Interestingly, miR-183 knockout mouse displays eye defects (Xiang et al., 2017), suggesting that absence of miR-182/miR-183 mediated regulation of Ago2 may contribute to these developmental defects.

One challenge of using animal models to determine the in vivo relevance of regulations of stemness is what phenotypes to look for in the genetically modified animals. The simplest scenario is the lethal phenotype, such as the *Ago2* knockout mouse. Complicated scenarios include animals that do not have lethal phenotype but with some defects in certain tissues or cells, such as the Lin28a knockout mouse (Sato et al., 2020). We believe using cell-based experimental system to generate mechanistic insights at the molecular level will be of great help and significance to interpret the phenotypes of animal models.

"Two lines of evidence indicated that miR-182/miR-183 targets Ago2 mRNA." – should use "regulates" instead of "targets", since the evidence for direct targeting comes later in the paper.

We made this change as suggested by the reviewer.

References:

Pulecio, J., Verma, N., Mejia-Ramirez, E., Huangfu, D., and Raya, A. (2017). CRISPR/Cas9-Based Engineering of the Epigenome. Cell Stem Cell *21*, 431-447.

Rajagopal, N., Srinivasan, S., Kooshesh, K., Guo, Y., Edwards, M.D., Banerjee, B., Syed, T., Emons, B.J., Gifford, D.K., and Sherwood, R.I. (2016). High-throughput mapping of regulatory DNA. Nat Biotechnol *34*, 167-174.

Sato, T., Kataoka, K., Ito, Y., Yokoyama, S., Inui, M., Mori, M., Takahashi, S., Akita, K., Takada, S., Ueno-Kudoh, H.*, et al.* (2020). Lin28a/let-7 pathway modulates the Hox code via Polycomb regulation during axial patterning in vertebrates. e*Life 9*.

Wang, X.W., Hao, J., Guo, W.T., Liao, L.Q., Huang, S.Y., Guo, X., Bao, X., Esteban, M.A., and Wang, Y. (2017). A DGCR8-Independent Stable MicroRNA Expression Strategy Reveals Important Functions of miR-290 and miR-183-182 Families in Mouse Embryonic Stem Cells. Stem Cell Reports *9*, 1618-1629.

Xiang, L., Chen, X.J., Wu, K.C., Zhang, C.J., Zhou, G.H., Lv, J.N., Sun, L.F., Cheng, F.F., Cai, X.B., and Jin, Z.B. (2017). miR-183/96 plays a pivotal regulatory role in mouse photoreceptor maturation and maintenance. Proc Natl Acad Sci U S A *114*, 6376-6381.

[Editors’ note: what follows is the authors’ response to the second round of review.]

Essential revisions:Reviewer #3:In this manuscript Liu and colleagues follow-up on their previous work showing that Ago2 expression is regulated in mESC by Trim-71 by proposing that Ago2 expression is also directly regulated by miR-182/miR-183 and that loss of this interaction affect mESC differentiation by resulting in let-7 stabilization.Using elegant genetic and biochemical approaches they convincingly show that Ago2 is directly repressed by miR-183/miR-183, as either loss of these two miRNAs or mutation of their binding sites in the Ago2 3'UTR results in upregulation of Ago2 expression by approximately 2 fold.They also show that targeted inactivation of miR-182/183 and, to a slightly lesser extent, mutation of miR-182/miR-183 binding sites in Ago2's 3'UTR promote differentiation of mESC in vitro.The manuscript is very well written, the experiments are elegant and include all the appropriate controls, and their results are largely consistent with the model proposed by the authors. While the authors convincingly demonstrate that miR-182/182 directly regulate Ago2, it is slightly less clear whether the proposed increased activity of let-7 can fully explain the observed phenotype.Overall this is a solid manuscript, although as discussed below not all conclusions are supported by conclusive evidence. I concur with reviewer 1 that the study provides relatively minor conceptual advance, but it can be argued that these results will be of interest to specialists in the field of miRNa and ESC biology.A major concern I have regards the strength of the conclusion that the impaired stem cell function upon loss of miR-182/183 activity on Ago2 is mediated by increased activity of let-7. The LNA-based experiments are suggestive, but although they prove that blocking let-7 helps preventing differentiation, they do not prove that increased let-7 function underlies the observed phenotype in the Ago2 UTR mutant. In fact, the only evidence that let-7 activity is increased is a modest upregulation of let-7 members expression. Whether this is sufficient to affect gene expression to a detectable extent is unclear. The evidence would be strengthened substantially if the authors showed increased let-7-mediated gene repression in the UTR mutant. A straightforward and relatively inexpensive experiment would be to perform RNAseq analysis on the wt and UTR-mutant mESCs. The authors' model leads to the prediction that the a relatively selective de-repression of predicted let-7 targets should be observed in the UTR mutant samples.

We appreciated the reviewer’s insightful comment!

We did exactly the same experiment as the reviewer described in our previous paper on the mESCs with the Trim71 binding site deleted in the 3’UTR of *Ago2* mRNA (Figure 4 in Liu et al., *eLife*, 2021). Through transcriptomic profiling via RNA-seq, we found that mRNAs with predicted let-7 binding sites showed increased level in the mutant mESCs than those in the WT mESCs compared to the mRNAs without let-7 binding sites, indicating increased let-7 miRNA activity (Figure 4E in Liu et al., *eLife*, 2021)

Here we used a different approach to address the same issue. Instead of examining mRNA levels, we measured protein level of an evolutionarily conserved let-7 target, Lin28a (the results are shown in Author response image 2). We observed that there was a decrease of Lin28a in the 3’UTR mutant mESCs. Moreover, there were no changes for Nanog and Oct4, which are not let-7 targets. There results argue that let-7 miRNA activity is increased in the 3’UTR mutant mESCs.

1) Figures 2C and 2D appear swapped in their description in the main text (lines 107-114)

We double checked Figures2C/2D and their corresponding descriptions in the main text, and we confirmed that they are in the correct order.

2) lines 132-135: I am not sure the authors can define the effect of the 3'UTR as not "in cis". Even if mediated by loss of miRNA binding, as is likely, the effect would still be on the transcript harboring the mutation, so technically, as far as I know, in cis. Perhaps a different language would be more appropriate. For example, the authors could say that the fact that deletion of miR-182/183 in a the 3'UTR-mut background doesn't result in additional upregulation of Ago2 provides additional evidence that the UTR mutation results in Ago2 upregulation by preventing miR-182/183 binding.

We made changes to correct the confusion caused by “in cis” in the revised manuscript (in line 132-134 of the revised manuscript).

3) It appears to me that loss of miR-182/183 has a stronger effect on ESC differentiation than mutation of their Ago2 3'UTR binding sites. This suggests that miR-182 promote ESC pluripotency by additional mechanism independent from Ago2. Perhaps the authors should comment on this in the discussion.

As the reviewer suggested, we added an additional paragraph (line 237-240) in the discussion commenting on this important implication.